# Psychometric Evaluation of the Nursing Outcome *Knowledge: Pain Management* in People with Chronic Pain

**DOI:** 10.3390/ijerph16234604

**Published:** 2019-11-21

**Authors:** Pedro Luis Pancorbo-Hidalgo, José Carlos Bellido-Vallejo

**Affiliations:** 1Department of Nursing, Faculty of Health Sciences, University of Jaén, Campus Las Lagunillas, s/n, 23071 Jaén, Spain; pancorbo@ujaen.es; 2Jaén University Hospital, Avenida del Ejército Español 10, 23007 Jaén, Spain

**Keywords:** chronic pain, *Knowledge: Pain Management*, nursing outcomes classification (NOC)

## Abstract

Pain has a major impact on health and quality of life. Since the level of knowledge of painful conditions can influence how these are addressed and managed, assessing this knowledge in patients becomes crucial. As a result, it is necessary to have culturally adapted and validated instruments that specifically measure patients’ knowledge of chronic pain management. The objective of this study was to carry out the Spanish cultural adaptation and the validation of the outcome *Knowledge: Pain Management* of the Nursing Outcomes Classification (NOC) in patients with chronic pain, defined as extent of understanding conveyed about causes, symptoms, and treatment of pain. A three-stage study was designed: (1) translation and cultural adaptation through an expert panel, (2) content validation, (3) clinical validation. This study provides nurses with a Spanish version of this scale adapted to their context, as well as a set of structured indicators to measure patients’ knowledge about chronic pain. The results indicated that the culturally adapted Spanish version of the outcome *Knowledge: Pain Management* had a high level of content validity (CVI = 0.92), with 27 indicators being distributed between two factors. This version has been shown to be reliable in terms of inter-observer agreement (κ = 0.79) and internal consistency (α = 0.95). In conclusion, *Knowledge: Pain Management* has been shown to be reliable and valid to measure knowledge of chronic pain.

## 1. Introduction

Pain is a health issue that affects a large number of individuals [1,2]. In acute conditions, pain is commonly temporary. However, when pain lasts longer than 3 months it is considered to be chronic pain and has an important impact on health and quality of life [3,4]. Pain is understood as a complex experience shaped by a wide variety of biological, psychological, and social factors [3]. This complex combination of factors includes past experiences of pain, age, education, culture, ethnicity, and gender, all of which endow the individual with certain beliefs and knowledge about pain [5,6].

Healthcare professionals’ knowledge of pain management and treatment has attracted considerable attention [7,8]. Conversely, patients’ knowledge of their own resources and experiences with pain has had considerably less attention [9]. There is a need to assess and address gaps in this respect, as a significant proportion of the general public report having a lack of knowledge of chronic pain [10], which can influence how chronic pain is addressed and managed.

In this sense, there is evidence that knowledge about analgesic drugs is associated with adherence to analgesic therapy [11]. Caregivers having better knowledge of pain management is not only associated with them having fewer concerns about pain and medications, but also with their patients experiencing less pain themselves [12]. In addition to treatment with medications, people may use different approaches to treating their pain; such as heat or cold application, relaxation techniques, exercise and physical activity [13,14] These therapies can improve patients’ functioning in activities of daily living [15].

An association has been found between the satisfaction with pain treatment and patients’ knowledge about the disease and some treatment-specific topics; this association persisted up to 12 months after receiving the education [16]. Participation in educational programs could improve pain management [17,18]. The ability of people to manage their pain increase when training on the nature and effects of pain, and how to live with pain, is provided [19]. Education must be accompanied by other approaches to pain management [19], addressing knowledge, skills, and attitudes jointly [20].

Improving patient empowerment requires, first, exploring their beliefs, level of understanding about pain and pain management. Beliefs about pain may influence the behavior of the patients in pain management [21]. Measuring the level of knowledge about pain is considered to be the first step before to propose strategies for help, reinforcement, or improvement. Some difficulties may arise when an individual makes decisions based on misconceptions founded on a lack of understanding and a faulty knowledge of pain. However, there seems to be no widespread consensus on how to measure the patients’ knowledge of pain. Kesänen et al. [22] conducted a systematic review in which they found that, of the 49 identified knowledge tests, few were oriented towards measuring pain knowledge [22]. Some of the instruments measuring knowledge do so only for a specific type of pain or population (like the Back Pain Knowledge Test) [23,24].

The Nursing Outcomes Classification (NOC) [25] includes the outcome *Knowledge: Pain Management* (KPM, code 1843), which is oriented towards measuring the extent of understanding conveyed about causes, symptoms, and treatment of pain. This outcome has 41 indicators (cognitive dimensions or aspects conveyed about causes, symptoms, and treatment of pain) which are used to measure the overall KPN concept. It is measured with a five-point Likert scale (1 = No knowledge, 2 = Limited knowledge, 3 = Moderate knowledge, 4 = Substantial knowledge, and 5 = Extensive knowledge). The indicators facilitate the monitoring of progress in knowledge of pain in a given episode of care and in different care settings [25].

The research of the NOC project were carried out in the United States. Subsequently, further studies have been carried out in other countries to increase the knowledge and applicability of the NOC outcomes, although each outcome has undergone its own research and development process. Validation by an expert panel has been used in certain nursing diagnoses [26,27], as well as correlation with other standards for assessing activities of daily living and functional status in nursing homes [28]. However, there are no known studies on the psychometric properties and cultural adaptation of KPM in countries other than the United States of America, despite this being considered particularly important in pain research [21,29].

The objective of this study was to provide the Spanish translation and cultural adaptation of the English version (5th edition) of the NOC outcome *Knowledge: Pain Management* and to assess its psychometric properties and sensitivity to change in patients with chronic pain.

## 2. Materials and Methods

A three-stage validation design was used for the outcome *Knowledge: Pain management*: (1) Spanish translation and cultural adaptation; (2) content validation; and (3) clinical validation.

Stage 1: The translation/back-translation method [29,30] was applied to the English version of the KPM (5th edition of the NOC) [31]. The label, definition, indicators, and measurement scale were translated into Spanish by a nurse with Spanish as her first language, who was an expert in nursing methodology. This translation was then compared with the Spanish versions of the KPM available in the 4th and 5th edition of the NOC [31,32,33]. A Spanish version was agreed upon and subsequently back-translated into English by a professional translator with English as his first language. This back-translation was submitted to the authors of the NOC at the Center for Nursing Classification & Effectiveness at the University of Iowa (Dr Swanson), who proposed a few modifications to the wording of some indicators. The first Spanish version was obtained as a result of this process.

Stage 2: A panel of 21 pain experts of Spanish nationality (19 nurses and two physiotherapists) assessed the content validity of the first Spanish version in two rounds of consensus using the Delphi technique [34]. The members of the panel had a mean amount of experience of 14.1 years in providing care to pain patients. Each expert independently scored the relevance of the indicators to the concept Knowledge about pain management, defined as the degree of understanding conveyed about causes, symptoms, and treatment of pain. The experts used a four-point Likert scale (1 = not appropriate, 2 = somewhat appropriate, 3 = moderately appropriate, 4 = completely appropriate), following the procedure proposed by Lynn [35] to calculate the content validity index (CVI). The same procedure was followed for the new dimensions proposed by the experts in the first round.

The CVI-indicator (CVI-i) was determined for each indicator, taking into consideration that, for six or more experts, this value should not be below 0.78 [35]. The modified kappa coefficient was also calculated for each indicator to assess the probability of random agreement among experts. For the overall score, the CVI-universal agreement (CVI-ua) and CVI-average (CVI-a) were calculated, considering CVI-a ≥ 0.80 to be an acceptable agreement [30,34,35].

Stage 3: An observational-longitudinal study was carried out to establish the reliability, validity, and sensitivity to change of the Spanish version of the KPM.

### 2.1. Setting

The study was conducted from March 2013 to April 2014 in 9 settings (primary care centers and hospitals) in the Andalusian Health Service in the provinces of Jaén (6) and Granada (3), Spain. Twenty nurses (16 women and four men) participated in the data collection. These nurses were working full time, had been working for more than 6 months in their respective healthcare centres, and had a mean experience of 20.7 years in caring for patients with chronic pain (Standard Deviation = 6.59). The nurses were working in the following units: Primary Care Team (50%), Oncology Impatient Unit (30%), Hospital Case Management (Advanced Practice Nurses) (10%), and Chronic Pain Unit (10%).

### 2.2. Sample

Data published by the authors of the validation of the NOC for use in people with chronic pain were used to estimate the size of the patient sample [36]. A minimum sample of 87 participants was deemed necessary assuming a kappa coefficient of 0.37, a 61% proportion of positive ratings for both observers, with a 95% confidence level. Patients were selected by non-probability sampling. Patients were invited to participate by collaborating nurses until a quota of 10 patients per centre was reached. The inclusion criteria were as follows: over 18 years of age; able to communicate in Spanish; and presence of chronic (cancer and non-cancer) pain for more than 6 months.

### 2.3. Main Research Variables

Knowledge: Pain management. The adapted Spanish version of KPM with 30 indicators, which was obtained after validation by experts, was used. These indicators measure the knowledge of how pain appears and is maintained, aspects related to medication and treatment, and various strategies and techniques for appropriate pain management. Twenty eight indicators were taken from the original English version [31] and a further two were proposed by the panel of experts: Multidimensional nature of pain (bio-psycho-social) and Benefits of physical exercise. The nurses assessed the patients and scored each indicator on a five-point Likert scale (from 1 = No knowledge to 5 = Extensive knowledge). Additionally, the option “Not applicable” was included, and the outcome Client satisfaction: Pain management was used to explore divergent validity.

Pain intensity. The Spanish version of the 11-point Numerical Pain Intensity Scale (NPIS) was used. The reliability and validity of this one-dimensional instrument are well-documented, being widely utilised for pain intensity assessment [37]. The aspects related to the experience of pain were measured using the Spanish version of a self-report questionnaire:

Pain Coping Questionnaire (reduced version) (PCQ-R). The subscale “Seeking instrumental social support” was considered. This subscale assesses how individuals cope with pain by seeking advice, help, or information (problem-focused coping) on a 5-point Likert scale (0 = Never, 1 = Sometimes, 2 = Often, 3 = Very often, and 4 = Always). The psychometric properties of this questionnaire have been studied in the Spanish population. Internal consistency ranged from α = 0.75 to α = 0.94; test-retest reliability was r = 0.81, and convergent validity with other scales was as expected [38,39].

Ad hoc questions to patients. Table 1 shows the five questions posed to patients about self-perception about pain.

### 2.4. Data Collection

The principal investigator trained all collaborating nurses in the measurement of the outcome *Knowledge: Pain Management* and in the data collection procedure. Most of the interviews with primary care patients were conducted in the nursing offices of the healthcare centres, although some were conducted in the patients’ homes. Inpatients were interviewed in a private area in the hospitalisation area. Patients in the Chronic Pain Unit were interviewed in the nurses’ office of the unit.

An ad hoc questionnaire were used by the nurses to record the patients’ demographic and clinical data, as well as their own demographic and professional data. Two assessments were conducted. In the first assessment, patients self-reported the NPIS, the PCQ-R, and the ad hoc questions about their knowledge and satisfaction. Two nurses then independently assessed the KPM, concealing their score from the other nurse (to measure inter-observer agreement). In the second assessment, conducted after a 20–30 day interval (to measure sensitivity to change), patients answered the ad hoc questions again and the KPM was reassessed by one of the nurses who conducted the first measurement.

### 2.5. Ethical Considerations

The project was approved by the Research Ethics Committee of Jaén, Spain. Nurses and patients were given detailed information on the project and signed an informed consent form. Participant anonymity and data confidentiality were preserved in compliance with current data protection laws in Spain.

### 2.6. Data Analysis

The data were collected in a spreadsheet (Microsoft Office Professional Plus 2013, Redmond, WA, USA) and analysed with SPSS (IBM^®^ SPSS^®^ statistics 19. IBM, International Business Machines Corporation, Armonk, NY, USA). Descriptive statistics were calculated. For nominal variables, frequencies and percentages were used. For continuous variables, means and standard deviations were used. Additionally, 95% confidence intervals were calculated for continuous variables.

An exploratory factor analysis (EFA) was performed using the principal components method with quartimax rotation to study the underlying conceptual structure of KPM. Sample adequacy was estimated with the Kaiser-Meyer-Olkin index and Bartlett’s test of sphericity, with *p* > 0.05 indicating that the model is appropriate. The number of factors was determined using the parallel analysis method following Horn’s criterion [40]. In the rotated factor matrix, indicators were considered to be saturated with values which were greater than 0.6 in one factor and with values which were smaller than 0.4 in the others.

Convergent and divergent validity was analysed using the multitrait-multimethod (MTMM) matrix [30,41]. In summary, Spearman’s correlations were calculated between the KPM score and the following: (a) the score for the outcome Client satisfaction: Pain management (same method, different trait), to explore divergent validity; (b) the score for the subscale “Seeking instrumental social support” from the PCQ-R (different method, different trait), to explore divergent validity; (c) the scores for the three satisfaction questions: How satisfied are you with the care provided by the nurses in relieving your pain?, How satisfied are you with the treatment of your pain?, and How much have you managed to improve your pain with the treatment and care provided by professionals? (different method, different trait), to explore divergent validity; and (d) the scores for the two questions about knowledge of pain management: How much do you know about the causes, symptoms, and treatment of pain? and How well prepared are you to take care of yourself and to manage and control your painful condition? (different method, same trait), to explore convergent validity.

In order to assess reliability, internal consistency and inter-observer agreement were determined. Cohen’s kappa coefficient with quadratic weighting was used to determine inter-observer agreement, considering kappa values ≤ 0.20 to reflect poor agreement; values of 0.21–0.40, mild agreement; 0.41–0.60, moderate agreement; 0.61–0.80, good agreement; and 0.81–1.00, excellent agreement. Internal consistency was calculated according to Cronbach’s alpha coefficient, considering values > 0.80 to indicate a good level of consistency [34,41].

Sensitivity to change. The capacity of KPM to detect change was analysed by administering this outcome at two points in time separated by 20–30 days (baseline and final). Wilcoxon’s rank-sum test was used to evaluate the mean agreement between the measurements, with statistical significance set at *p* < 0.05.

## 3. Results

### 3.1. Stage 1. Translation and Cultural Adaptation

A Spanish adaptation of KPM, semantically equivalent to the English version, was obtained. The adapted version has modifications to the label, to 17 indicators, and to the measurement scale, compared to the 5th and 6th Spanish editions of the NOC [31,32,33].

### 3.2. Stage 2. Content Validation

After review by the panel of experts, a version with 30 indicators was obtained (28 from the original version; a further two were added). Overall, the CVI-ua was 0.33 and the CVI-a was 0.92. All indicators were rated as excellent with modified kappa scores ranging from 0.75 to 1.00 (Table 2). Thirteen indicators from the original version and two new indicators were eliminated due to the low agreement among the experts (with modified kappa scores ranging from 0.27 to 0.77).

### 3.3. Stage 3. Clinical Test

#### 3.3.1. Patients’ Characteristics

The clinical validation study included 84 pain patients. Table 3 shows the patients’ clinical and demographic characteristics.

The most frequent nursing diagnoses (NANDA-I) were Chronic pain (42.9%) and Acute pain (11.9%). Patients were using different types of drugs (mean = 3.15; SD = 1.43), with scheduled non-opioid analgesics being more frequent (mean = 1.21; SD = 0.81). All 84 patients underwent a first (baseline) measurement (*n* = 84), and 44 patients had a second measurement (final). The mean interval between the initial and final measurements was 42.84 (SD = 19.35) days.

#### 3.3.2. Pain Assessment

Pain intensity, measured with the NPIS by the patients themselves when completing the self-report questionnaires, was 5.37 (SD = 2.20). Pain was mainly located in the back (54.8%), lower extremities (48.8%), and pelvis (45.2%). The mean score obtained in the PCQ-R subscale “Seeking instrumental social support” was 8.73 points (SD = 4.29) (range: 0–16) (Table 3).

The overall mean score obtained through the outcome *Knowledge: Pain Management* was 3.24 (SD = 0.87) in the baseline measurement and 3.46 (SD = 0.77) in the final measurement. The lowest mean score was 2.52 (SD = 1.31) for the indicator “Potential medication interactions”, and the highest mean score was 3.98 (SD = 0.88) for “Importance of informing health professional of all current medication”. The scores for the 5 questions to patients about their knowledge and satisfaction with pain management are shown in Table 4. There is a slight improvement in the two questions about patients’ knowledge and perceived readiness to manage this pain, but no improvement in the three questions about satisfaction with care.

#### 3.3.3. Psychometric Assessment of the Spanish Adaptation of KPM

Construct validity was assessed through an EFA. Taking into consideration the parallel analysis, the 30 indicators can be grouped into two factors. The Kaiser-Meyer-Olkin sample adequacy test (0.806) and Bartlett’s test of sphericity (1770.63; *p* < 0.0001) showed that the data were suitable for analysis.

A preliminary EFA showed that three indicators, i.e., Safe use of prescribed medication (variance = 0.496), Effective heat/cold application (variance = 0.353), and Available community resources (variance = 0.279), presented low proportions of explained variance and low factor loading values. As a result, they were eliminated.

A second analysis with quartimax rotation showed a structure of 27 indicators distributed between two factors explaining 53.56% of the variance. The matrix of rotated factors shows their composition: (a) Knowledge about pain, made up of 25 indicators explaining 45.73% of the variance; (b) Non-prescription medication, made up of two indicators explaining 7.82% of the remaining variance (Table 5).

#### 3.3.4. Multitrait-Multimethod Matrix

Analysis using MTMM makes it possible to assess the convergent and divergent validity of KPM (Table 6). For convergent validity, the KPM outcome score was moderately correlated with the two ad-hoc questions on patient knowledge: Q1 How much do you know about the causes, symptoms, and treatment of pain? (r = 0.40; *p* < 0.001); and Q2 How well prepared are you to take care of yourself and to manage and control your painful condition? (r = 0.39; *p* < 0.001).

For divergent validity, weak correlations lacking statistical significance were found with the score for the outcome Client satisfaction: Pain management (r = 0.16; *p* = 0.16) and with the PCQ-R subscale “Seeking instrumental social support” (r = 0.18; *p* = 0.11); and no correlations with the three questions about satisfaction: Q3 How satisfied are you with the care provided by the nurses in relieving your pain? (r = 0.04; *p* = 0.70); Q4 How satisfied are you with the treatment of your pain? (r = 0.05; *p* = 0.68); and Q5 How much have you managed to improve your pain with the treatment and care provided by professionals? (r = 0.05; *p* = 0.63).

#### 3.3.5. Reliability

The overall KPM score showed good inter-observer agreement (kappa = 0.79; 95% CI = 0.68–0.90; *p* < 0.001). Agreement was excellent for 6 indicators and good for 20 indicators (Table 6). The internal consistency values were α = 0.95 for the final 27-indicator version, α = 0.95 for the factor Knowledge about pain, and α = 0.94 for Non-prescription medication (Table 7).

#### 3.3.6. Sensitivity to Change

No differences were found between the baseline and final scores for the outcome KPM (baseline score = 3.49; final score = 3.46 (Z = −0.68; *p* = 0.50)), nor for either of the two ad-hoc questions: How much do you know about the causes, symptoms, and treatment of pain? (Wilcoxon’s Z = −1.42; *p* = 0.15) and How well prepared are you to take care of yourself and to manage and control your painful condition? (Wilcoxon’s Z = −1.04; *p* = 0.29). However, in the analysis of indicators, statistically significant differences were observed in six of them between the baseline and final measurements: Strategies for preventive pain management (Z = 2.42; *p* = 0.01), Effective relaxation techniques (Z = 2.14; *p* = 0.03), Effective distraction (Z = 2.00; *p* = 0.046), Effective heat/cold application (Z = 2.40; *p* = 0.02), Benefits of ongoing self-monitoring of pain (Z = 2.43; *p* = 0.01), Available community resources (Z = 2.10; *p* = 0.03).

## 4. Discussion

This article report on the process to translate and adapt to Spanish context and culture the set of indicators included in the nursing outcome *Knowledge: Pain Management*. The psychometric properties (reliability and validity) of this outcome were tested in a sample of patients with pain. Nurses in Spanish-speaking contexts could use this adapted version of KPM to measure the degree of knowledge that their patients have about the causes, symptoms, and treatment of pain.

### 4.1. Cultural Adaptation

Following the principles of good practice [29,42], during the process of cultural adaptation we worked on the semantic and conceptual equivalence not only of the label and definition of the outcome, but also of its indicators and measurement scale. Since the KPM indicators are the specific elements that measure particular aspects of knowledge, their adaptation is especially relevant so that patients are able to understand what information is being requested from them and so that they can convey precise answers, and professionals are able to distinguish between ignorance and knowledge in the different dimensions that make up the KPM outcome. On the basis of the data from this study and other similar studies [43,44], this translation and cultural adaptation process results in a clearer and more comprehensible wording of the outcome definition and its indicators than that obtained by directly translating it from English into Spanish without expert assessment.

### 4.2. Content Validity

The content validity data suggest that the set of indicators is adequate for the conceptualisation of the KPM outcome in accordance with the recommendations made by Polit et al. [30], who proposed that CVI-a values greater than 0.90 indicate excellent content validity. The elimination of 13 indicators is a substantial reduction from the original version. Almost a third of the dimensions did not adequately represent the overall concept according to the group of experts, who even considered that some of them were not applicable in the care setting of this study, e.g., indicators Benefits of hypnosis (CVI-i = 0.27) and Benefits of biofeedback (CVI-i = 0.30), quite uncommon in the Spanish context. This invites reflection for two separate reasons. First, the KPM outcome has to contain the essential indicators that define it and adequately represent it to provide nurses with an operative list for their clinical work (the extensive initial list had 41 indicators). Secondly, it is necessary to conduct new content validity studies in other cultural settings to verify the content validity indices and the inclusion or exclusion of indicators resulting from this study.

### 4.3. Construct Validity

The two-factor structure proposed in the final model for the KPM outcome makes it possible to explain a significant percentage of the overall variance, although a fraction of the variance remains unexplained, so it is likely that other indicators may be added in the future. Further studies are required to verify the suitability of this structure in other clinical settings and types of pain patients. Factor 1, Knowledge about pain, includes a considerable number of indicators addressing various concepts of pain, such as treatment and improvement strategies (13), aspects related to medication (9), and, to a lesser extent, signs and symptoms associated with pain (3), which point to the complexity of this factor. Factor 2, Non-prescription medication, has only two indicators. The correlations of these indicators with the total are the lowest on the whole list, which suggests that factor 2 needs an improved definition of the overall set and/or its indicators. This distribution of outcome indicators into two factors may be useful for nurses when deciding which indicators to use in assessing patients’ knowledge of different aspects related to pain and pain management, in line with other validation studies of other NOC outcomes [43,44,45].

### 4.4. Criterion Validity (Convergent and Divergent)

We found evidence of validity of convergent and divergent criteria for the KPM outcome. Convergent validity was verified drawing on the strong correlations with the specific questions about knowledge and ability to control pain, as well as on each of the two proposed factors. In addition, the KPM outcome is well differentiated from other related yet different concepts, such as satisfaction with pain management and ability to cope, as shown by the divergent validity analysis.

A more detailed analysis showed that some of the indicators referring to medication aspects had a weak correlation with the score on the ad hoc question about knowledge. At this point, it is pertinent to wonder whether the question was well configured in terms of content and scope for the population studied, as this suggests that knowledge about medication can be challenging for patients. The level of knowledge is associated with adherence to analgesic therapy [11], but knowledge may be insufficient by itself, as adherence may be influenced by other aspects such as motivation and behavioural skills [46], or by patients’ attitudes and beliefs about medication [47].

### 4.5. Reliability

We found evidence of the reliability of KPM when used by nurses in patients with chronic pain. Inter-observer agreement was high for the overall set and most of the indicators. However, five indicators relating to medication had a low or moderate correlation [34], which may indicate that medication-related issues also pose challenges to nurses [48,49]. Internal consistency was high and there was adequate correlation between the total and each of the indicators, with no scores suggesting that any indicator should be eliminated [41,50].

### 4.6. Sensitivity to Change

It could not be established that KPM showed sensitivity to changes in patient knowledge in the overall score. Patients may not have experienced substantial changes in their knowledge which would explain why no significant changes were observed between the baseline and final scores. However, significant changes were observed in a group of indicators relating to various pain management and treatment strategies. In this study, the mean time interval between measurements was almost 43 days, longer than the interval in the study on the outcome Pain control (code 1605) [44], with 30-day intervals between measurements. Increasing the time between measurements may be the key to revealing sensitivity in populations of long-term pain patients. Studies involving an educational intervention may also provide further evidence on the sensitivity to change of the KPM outcome.

Although no validation studies have been identified with which to compare the psychometric tests in this study, the usefulness of KPM has been demonstrated in other studies. As such, KPM has been validated for certain nursing diagnoses, Lucena et al. [26] validated KPM using a group of experts for the following nursing diagnoses: Risk for Frail Elderly Syndrome and Frail Elderly Syndrome. KPM was included in the care plan of a patient with long-term left ventricular assist device (associated with the nursing diagnosis Acute pain) [27]. This approach to patients with acute (short-term) pain coincides with the small sample of patients in this study who presented with pain for more than one month. KPM was also considered useful in evaluating the evidence of quality of care provided by nurses in nursing homes in relation to the administrative rules measuring quality of care in the United States of America [28].

### 4.7. Limitations

A comprehension and cognitive equivalence test carried out with nurses, patients, healthy individuals, and even native speakers representing the target population (sex, age, level of education, diagnosis) may improve the cultural adaptation process and provide a better adjustment of the concepts used to the cultural characteristics of the given geographical area [42,51].

Homogeneity in the training of participating nurses cannot be ensured, as they were included in the study because they were willing to collaborate and this may, therefore, lead to an overestimation of reliability. Attempts have been made to control bias by training participants in the conditions of the study, by diversifying participation (from clinical settings to rural and urban health centres, hospitals, and specialised chronic pain units), and by seeking the participation of advanced practice nurses [52].

Patients were selected using convenience sampling and the sample size was relatively small. However, the representativeness of the study was improved by sampling at 9 different locations. Broader randomised samples and other additional care settings should be included to further increase knowledge of the psychometric properties of KPM. Some of the indicators included in the English version of this outcome were not included in the two-factor Spanish version, which may require further assessments. Additional studies are also necessary to increase the time between the baseline and the final measurements, seeking significant differences in the sensitivity to change of KPM and its indicators in patients with chronic pain.

## 5. Conclusions

The final version of the nursing outcome *Knowledge: Pain Management*, adapted to a Spanish context, includes 27 indicators. Evidence has been obtained of the reliability and validity of these indicators for use in nursing practice in Spanish-speaking contexts.

An internal structure of two factors, referred to as Knowledge about pain and Non-prescription medication, can be observed. This structure is novel and thus requires further research with larger patient samples, additional care settings, and acute pain scenarios. Being able to make an appropriate selection of indicators based on this two-factor structure may help nurses in their decision-making, especially the less experienced.

The present study highlights an interesting field of research. Improving the definition and conceptual configuration of the outcome KPM, presenting an operational list of indicators, and increasing the input of psychometric testing through further research will not only consolidate the reliability and validity of this outcome, but will also make it more useful in the complex process of caring for individuals with pain.

## Figures and Tables

**Table 1 ijerph-16-04604-t001:** Questions to patients about their pain.

Question	1	2	3	4	5
How much do you know about the causes, symptoms, and treatment of pain?	Nothing	Very little	Neither little nor much	Quite	A lot
How well prepared are you to take care of yourself and to manage and control your painful condition?	Not at all prepared	Very little prepared	Neither little nor much prepared	Quite prepared	Very prepared
How satisfied are you with the care provided by the nurses in relieving your pain?	Not at all satisfied	Somewhat satisfied	Moderately satisfied	Very satisfied	Completely satisfied
How satisfied are you with the treatment of your pain?	Not at all satisfied	Somewhat satisfied	Moderately satisfied	Very satisfied	Completely satisfied
How much have you managed to improve your pain with the treatment and care provided by professionals?	Nothing	Very little	Moderately	Quite	Completely

**Table 2 ijerph-16-04604-t002:** Content validity index and ratings of the indicators of the outcome *Knowledge: Pain Management.*

Indicators ^a^		Scores of 3 or 4/N of Experts ^b^	CVI-i ^c^	Pa ^d^	K* ^e^	Rating ^f^
184301	Causes and contributing factors of pain	18/18	1	0	1	Excellent
184302	Signs and symptoms of pain	17/18	0.94	0	0.94	Excellent
184303	Strategies to control pain	18/18	1	0	1	Excellent
184304	Strategies to manage chronic pain	17/18	0.94	0	0.94	Excellent
184305	Prescribed medication regimen	18/18	1	0	1	Excellent
184306	Correct use of prescribed medication	18/18	1	0	1	Excellent
184307	Correct use of non-prescription medication	18/18	1	0	1	Excellent
184308	Safe use of prescribed medication	17/18	0.94	0	0.94	Excellent
184309	Safe use of non-prescription medication	16/18	0.89	0	0.89	Excellent
184310	Medication therapeutic effects	18/18	1	0	1	Excellent
184311	Medication side effects	15/17	0.88	0	0.88	Excellent
184312	Medication adverse effects	17/18	0.94	0	0.94	Excellent
184313	Potential medication interactions	15/18	0.83	0	0.83	Excellent
184315	Safety issues related to medication	17/18	0.94	0	0.94	Excellent
184318	Importance of complying with medication regimen	18/18	1	0	1	Excellent
184319	Importance of complying with medication regimen	18/18	1	0	1	Excellent
184320	Activity restrictions	17/18	0.94	0	0.94	Excellent
184321	Activity precautions	15/18	0.83	0	0.83	Excellent
184322	Effective positioning techniques	16/18	0.89	0	0.89	Excellent
184323	Effective relaxation techniques	15/18	0.83	0	0.83	Excellent
184325	Effective distraction	14/17	0.82	0.01	0.82	Excellent
184326	Effective heat/cold application	15/18	0.83	0	0.83	Excellent
184334	Benefits of ongoing self-monitoring of pain	16/18	0.89	0	0.89	Excellent
184335	Benefits of lifestyle modifications to reduce pain	18/18	1	0	1	Excellent
184337	Strategies for preventive pain management	17/18	0.94	0	0.94	Excellent
184338	When to obtain assistance from a health professional	18/18	1	0	1	Excellent
184340	Available community resources	15/18	0.83	0	0.83	Excellent
184341	Reputable sources of information	15/17	0.88	0	0.88	Excellent
New	Multidimensional nature of pain (bio-psycho-social)	19/21	0.90	0	0.90	Excellent
New	Benefits of physical exercise	16/21	0.76	0.01	0.75	Excellent
	IVC-universal agreement		0.33		0.33	
	IVC-average ^h^		0.92		0.92	

^a^ Indicators included in the clinical version after the second consensus round. ^b^ Degree to which indicators were appropriate for the outcome *Knowledge: Pain Management*; Indicators with expert scores of 3 (moderately appropriate) or 4 (completely appropriate). ^c^ CVI-i = Content Validity Index-indicator (Number of experts giving scores of 3 or 4/Number of experts giving scores). ^d^ Pa. (Probability of random agreement among experts) = ([N!/A!/(N-A)!]*0.5 N), where N = number of experts and A = agreement on relevance (score of 3 or 4). ^e^ K* = modified kappa. Content validity index after correcting for chance agreement. k* = (CVI-i − Pa)/(1−Pa). ^f^ Rating. k evaluation criteria: k ≤ 0.39 = poor; k 0.40-0.59 = moderate; k 0.60-0.74 = good; k > 0.74 = excellent.

**Table 3 ijerph-16-04604-t003:** Characteristics of patients assessed with *Knowledge: Pain management* (*N* = 84).

Variables	Frequency (%)	
SEX		
Female	48 (57.1)	
Male	36 (42.9)	
Marital status		
Married	62 (73.8)	
Single	13 (15.5)	
Divorced	5 (6.0)	
Widow/er	4 (4.8)	
Level of education		
Primary education	43 (51.2)	
No education	21 (25.0)	
Secondary education	8 (9.5)	
Higher education/University education	6 (7.1)	
Higher non-university education	3 (3.6)	
Not stated	3 (3.6)	
Employment status		
Retired	40 (47.6)	
Household tasks	19 (22.6)	
Unemployed	15 (17.9)	
Employed	9 (10.7)	
Not stated	1 (1.2)	
How long have you been in pain?		
> 3 years	38 (45.2)	
> 1 year	15 (17.9)	
> 6 months	11 (13.1)	
> 3 months	12 (14.3)	
> 1 months	7 (8.3)	
Not stated	1 (1.2)	
Location of pain		
Back	46 (54.8)	
Lower limbs	41 (48.8)	
Pelvis	38 (45.2)	
Neck	31 (36.9)	
Upper extremities	26 (31.0)	
Abdomen	24 (28.6)	
Head/face	17 (20.2)	
Chest	17 (20.2)	
Setting		
Primary care	40 (47.6)	
Hospital	36 (42.8)	
Chronic pain unit	8 (9.5)	
	Mean (SD)	Range
AGE (years)	55.49 (14.11)	18–81
Baseline pain level (NPIS)	5.37 (2.20)	0–10
Pain coping questionnaire	8.73 (4.29)	0–16

**Table 4 ijerph-16-04604-t004:** Ad hoc questions about knowledge and satisfaction. Patient response scores.

Questions	Mean Score (SD)	
	Baseline *	Final *
How much do you know about the causes, symptoms, and treatment of pain?	3.06 (0.91)	3.32 (0.77)
How well prepared are you to take care of yourself and to manage and control your painful condition?	2.92 (1.02)	3.36 (0.81)
How satisfied are you with the care provided by the nurses in relieving your pain?	3.90 (0.91)	3.55 (1.15)
How satisfied are you with the treatment of your pain?	3.37 (1.02)	3.11 (1.02)
How much have you managed to improve your pain with the treatment and care provided by professionals?	3.53 (0.89)	3.09 (0.88)

* Range: from 1 to 5.

**Table 5 ijerph-16-04604-t005:** Construct validity (principal component analysis with quartimax rotation) and criterion validity for the outcome *Knowledge: Pain Management.*

Knowledge: Pain Management		Factors		Ad-hoc Question A ^a^	Ad-hoc Question B ^a^	Score ^b^
		1	2	Correlation (ρ) ^c^		
Overall score. Final version (27 indicators) ^b^				0.39 *	0.44 *	
Factor 1: Knowledge about pain ^d^				0.39 *	0.45 *	0.99 *
184312	Medication adverse effects	0.80	−0.34	0.36 *	0.26 *	0.70 *
184313	Potential medication interactions	0.80	−0.24	0.34 *	0.28 *	0.71 *
184315	Safety issues related to medication	0.78	−0.22	0.40 *	0.31 *	0.72 *
184311	Medication side effects	0.76	−0.23	0.29 *	0.26 *	0.71 *
184323	Effective relaxation techniques	0.75	−0.20	0.34 *	0.37 *	0.67 *
184335	Benefits of lifestyle modifications to reduce pain	0.73	0	0.28 *	0.33 *	0.67 *
184318	Importance of complying with medication regimen	0.73	0	0.27 *	0.32 *	0.70 *
184301	Causes and contributing factors of pain	0.72	−0.12	0.50 *	0.52 *	0.60 *
184302	Signs and symptoms of pain	0.72	0.24	0.38 *	0.50 *	0.67 *
184320	Activity restrictions	0.71	0.33	0.25 *	0.30 *	0.68 *
New	Multidimensional nature of pain (bio-psycho-social)	0.71	0.12	0.34 *	0.50 *	0.64 *
184303	Strategies to control pain	0.71	0.45	0.26 *	0.31 *	0.64 *
184337	Strategies for preventive pain management	0.70	0	0.23 *	0.31 *	0.65 *
184310	Medication therapeutic effects	0.69	010	0.17	0.24 *	0.63 *
184304	Strategies to manage chronic pain	0.68	0.37	0.20	0.26 *	0.65 *
184321	Activity precautions	0.68	0.26	0.28 *	0.28 *	0.66 *
184305	Prescribed medication regimen	0.66		0.20	0.39 *	0.64 *
184322	Effective positioning techniques	0.64	0.15	0.32 *	0.35 *	0.65 *
184306	Correct use of prescribed medication	0.64	0	0.13	0.28 *	0.64 *
184334	Benefits of ongoing self-monitoring of pain	0.63	0	0.28 *	0.18	0.61 *
New	Benefits of physical exercise	0.63		0.28 *	0.41 *	0.65 *
184325	Effective distraction	0.63	−0.19	0.29 *	0.42 *	0.58 *
184338	When to obtain assistance from a health professional	0.63	0	0.32 *	0.25 *	0.67 *
184319	Importance of complying with medication regimen	0.63	0.23	0.26 *	0.28 *	0.67 *
184341	Reputable sources of information	0.60	0	0.38 *	0.26 *	0.60 *
Factor 2: Non-prescription medication ^d^				0.30 *	0.27 *	0.65 *
184309	Safe use of non-prescription medication	0.37	0.74	0.17	0.24 *	0.36 *
184307	Correct use of non-prescription medication	0.32	0.70	0.12	0.19	0.34 *
Eigenvalues		12.35	2.11			
Percentage of variance explained		45.73	7.82			
Accumulated percentage of variance explained		48.73	53.56			

In bold: values with higher scores in one of the two factors of the rotated factors matrix. ^a^ Ad-hoc question. (A) How much do you know about the causes, symptoms, and treatment of pain? (B) How well prepared are you to take care of yourself and to manage and control your painful condition? ^b^ Version with 27 indicators. Version after principal component analysis, structured into two factors. Correlation of each factor/indicator with the overall score of the final version. ^c^ Rho. Spearman’s correlation coefficient. ^d^ Factors. The mean score for each factor was correlated with the score for the response to the ad-hoc question, as well as with the mean score for the final version. * Statistically significant value (*p* < 0.05).

**Table 6 ijerph-16-04604-t006:** Matrix of correlations between the scores obtained for the outcome *Knowledge: Pain management* and the scores for other instruments and ad-hoc questions (multitrait-multimethod matrix).

Instruments	Title	NOC Outcomes		Questionnaire PCQ-R	Questions about Knowledge		Questions about Satisfaction	
		**KPM**	**CSPM**		**Q1**	**Q2**	**Q3**	**Q4**
NOC outcomes	KPM							
	CSPM	0.16						
Questionnaire	PCQ-R	0.18	0.03					
Questions about knowledge	Q1	0.40 *	0.17	0.32 *				
	Q2	0.39 *	−0.05	0.15	0.55 *			
Questions about satisfaction	Q3	0.04	0.29 *	0.45 *	−0.04	0.10		
	Q4	−0.04	0.50 *	0.20	0.11	0.05	0.46 *	
	Q5	0.05	0.46 *	0.15	0.12	0.13	0.37 *	0.59 *

NOC outcomes: KPM = *Knowledge: Pain Management*; CSPM= *Client satisfaction: Pain management*. PCQ-R questionnaire = Pain Coping Questionnaire (reduced version). Questions: Q1 = How much do you know about the causes, symptoms, and treatment of pain?; Q2 = How well prepared are you to take care of yourself and to manage and control your painful condition?; Q3 = How satisfied are you with the care provided by the nurses in relieving your pain?; Q4 = How satisfied are you with the treatment of your pain?; Q5 = How much have you managed to improve your pain with the treatment and care provided by professionals?

**Table 7 ijerph-16-04604-t007:** Reliability of the outcome *Knowledge: Pain management*. Inter-observer agreement and internal consistency.

Knowledge: Pain Management		Initial Assessment	Inter-Observer Agreement			Internal Consistency	
		Mean (SD)	*N*	KAPPA	(95%) CI	Correlation Item-Total	*α*
Overall scores (Baseline Version) ^a^		3.24 (0.87)	78	0.79 *	0.68 0.90		0.95
Factor 1: Knowledge about pain							0.95
184312	Medication adverse effects	2.80 (1.33)	84	0.66 *	0.51 0.81	0.72	
184313	Potential medication interactions	2.52 (1.31)	83	0.75 *	0.64 0.82	0.73	
184315	Safety issues related to medication	2.94 (1.26)	83	0.81 *	0.73 0.89	0.74	
184311	Medication side effects	2.94 (1.29)	82	0.72 *	0.59 0.84	0.68	
184323	Effective relaxation techniques	2.57 (1.33)	82	0.78 *	0.68 0.87	0.68	
184335	Benefits of lifestyle modifications to reduce pain	3.08 (1.16)	83	0.81 *	0.72 0.89	0.71	
184318	Importance of complying with medication regimen	3.76 (1.04)	83	0.64 *	0.49 0.78	0.69	
184301	Causes and contributing factors of pain	3.23 (1.20)	84	0.85 *	0.79 0.92	0.66	
184302	Signs and symptoms of pain	3.58 (0.92)	84	0.77 *	0.69 0.86	0.70	
184320	Activity restrictions	3.64 (1.04)	83	0.67 *	0.52 0.82	0.66	
New	Multidimensional nature of pain (bio-psycho-social)	3.34 (1.06)	83	0.79 *	0.70 0.88	0.67	
184303	Strategies to control pain	3.38 (1.02)	81	0.79 *	0.68 0.89	0.69	
184337	Strategies for preventive pain management	3.04 (1.17)	84	0.89 *	0.83 0.95	0.63	
184310	Medication therapeutic effects	3.42 (1.11)	84	0.58 *	0.39 0.77	0.66	
184304	Strategies to manage chronic pain	3.30 (1.02)	84	0.70 *	0.57 0.82	0.67	
184321	Activity precautions	3.59 (1.05)	84	0.73 *	0.60 0.86	0.62	
184305	Prescribed medication regimen	3.82 (1.00)	83	0.70 *	0.58 0.82	0.62	
184322	Effective positioning techniques	3.36 (1.15)	83	0.77 *	0.66 0.88	0.60	
184306	Correct use of prescribed medication	3.83 (1.01)	84	0.69 *	0.58 0.80	0.62	
184334	Benefits of ongoing self-monitoring of pain	2.82 (1.15)	82	0.85 *	0.75 0.93	0.60	
New	Benefits of physical exercise	3.16 (1.22)	82	0.79 *	0.71 0.87	0.60	
184325	Effective distraction	2.84 (1.16)	82	0.85 *	0.79 0.91	0.58	
184338	When to obtain assistance from a health professional	3.36 (1.20)	81	0.72 *	0.62 0.82	0.59	
184319	Importance of complying with medication regimen	3.98 (0.88)	79	0.61 *	0.44 0.78	0.60	
184341	Reputable sources of information	2.98 (1.23)	83	0.74 *	0.61 0.86	0.56	
Factor 2: Non-prescription medication							0.93
184309	Safe use of non-prescription medication	3.45 (1.24)	83	0.36 *	0.11 0.60	0.45	
184307	Correct use of non-prescription medication	3.50 (1.21)	84	0.53 *	0.31 0.76	0.39	
**Indicators not included**							
184308	Safe use of prescribed medication	3.78 (1.00)	84	0.52 *	0.28 0.75	0.58	
184326	Effective heat/cold application	2.77 (1.27)	82	0.74 *	0.58 0.90	0.52	
184340	Available community resources	2.86 (1.21)	83	0.66 *	0.48 0.84	0.40	

^a^ Overall score = the clinical judgment of nurses on KPM (30 indicators). * Statistically significant inter-observer correlation values (*p* < 0.05). Range: 1 to 5 for all indicators.

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
