# Peer review of "Psychometric Evaluation of the Nursing Outcome Knowledge: Pain Management in People with Chronic Pain"

_ijerph, 2019, doi:10.3390/ijerph16234604_

Round 1

Reviewer 1 Report

Thank you for the opportunity to review this paper that describes the validation process of the Nursing outcomes classification outcome for knowledge: pain management (Code 1843). While the article provides sequential detail regarding the process, other fundamental information is lacking and required remediation to enable understanding by the reader.

The abstract is unclear about what the Spanish cultural adaptation was achieving.  The abbreviation 'NOC' is not explained until P2L62. The italicised 'knowledge: pain management' is not described or defined. Readers may not be familiar with this outcome, let alone the need for a Spanish validation process. The final line in the abstract needs to be much further up in the paragraph.

Terminology within the article needs to be carefully scrutinised.  The word 'problem' P1L29 could be changed to 'issue' and the word 'enjoyed' probably needs to be exchanged for a more appropriate term such as 'had'.

The manuscript has a few very long and complex sentences ie P1L43 onwards; P2L50 onwards; and the conclusion is comprised of a series of long sentences that need to be simplified. Please check and simplify these examples and proof read for others.

Some sentences seem to be in reverse order eg P2L45; P2L49, the important parts of the sentence are placed second, rather than first.  Changing the order would improve clarity and readability.

Removal of the extraneous words that do not add to the content ie P2L49 'therefore' is unnecessary; P2L68 'or lack of', remove; P2L73 'In the case of'; P3L98 'to this end'. 

P2L86 and 89 - instead of 'mother tongue' which is colloquial the term 'first language' could be used.

There is also a mix of tenses that reduces readability ie P2 line 50 onwards; P4L178 onwards... swaps between present and past tense. This interchange occurs throughout the manuscript and needs attention to improve readability. 

P3L100, the reference for Lynn needs to follow directly after, rather than the end of the sentence.

Consistency with reference to the 'Knowledge:pain management' in italics or not would improve readability.

Would a table to describe P4L148 improve readability? The tables presented in the word columns ie Table 1 indicators; Table 2 variables; Table 3 questions; Table 4 overall score; Table 5 no heading but content below; Table 6 overall score  need to be left justified to improve readability. 

In 2.6, precise detail regarding what spreadsheet was used and what version of SPSS as per standard reporting would be useful.

The first paragraph of the discussion P11L317 onwards is repetitious.

The process of validation provides logical sequencing, however, due to the presentation of the tables, it was hard to read and comprehend. I suggest the journal table specifications are checked and complied with to improve readability.

Author Response

Response to Reviewer 1 Comments

Point 1: Thank you for the opportunity to review this paper that describes the validation process of the Nursing outcomes classification outcome for knowledge: pain management (Code 1843). While the article provides sequential detail regarding the process, other fundamental information is lacking and required remediation to enable understanding by the reader.

Response 1: Thank you for these detailed comments that have helped us to improve the article.

Point 2: The abstract is unclear about what the Spanish cultural adaptation was achieving.  The abbreviation 'NOC' is not explained until P2L62. The italicised 'knowledge: pain management' is not described or defined. Readers may not be familiar with this outcome, let alone the need for a Spanish validation process. The final line in the abstract needs to be much further up in the paragraph.

Response 2: We have modified the Abstract according these suggestions. We have included an explanation of the cultural adaptation; the meaning of NOC; defining the term ‘Knowledge: pain management’ and slightly changing the structure of the abstract.

Knowledge: Pain management is the name proposed in the Nursing Outcomes Classification to refer to this concept. We have put the name in italics through all the article.

Point 3: Terminology within the article needs to be carefully scrutinised.  The word 'problem' P1L29 could be changed to 'issue' and the word 'enjoyed' probably needs to be exchanged for a more appropriate term such as 'had'.

Response 3: We have made these corrections, as suggested.

Point 4: The manuscript has a few very long and complex sentences ie P1L43 onwards; P2L50 onwards; and the conclusion is comprised of a series of long sentences that need to be simplified. Please check and simplify these examples and proof read for others.

Response 4: We have revised these sentences and put them in short and clearer sentences. Thank you for the observation.

Point 5: Some sentences seem to be in reverse order eg P2L45; P2L49, the important parts of the sentence are placed second, rather than first.  Changing the order would improve clarity and readability.

Response 5: These sentences have been revised and re-written changing the order of some words.

Point 6: Removal of the extraneous words that do not add to the content ie P2L49 'therefore' is unnecessary; P2L68 'or lack of', remove; P2L73 'In the case of'; P3L98 'to this end'. P2L86 and 89 - instead of 'mother tongue' which is colloquial the term 'first language' could be used.

Response 6: We have revised all these paragraphs to remove or change some extraneous words.

Point 7: There is also a mix of tenses that reduces readability ie P2 line 50 onwards; P4L178 onwards... swaps between present and past tense. This interchange occurs throughout the manuscript and needs attention to improve readability. 

Response 7: We have modified these sentences and changed the tense or verbs. We have used past tense when referring to results of previous studies; but the present tense for making statements for established facts or knowledge.

Point 8: P3L100, the reference for Lynn needs to follow directly after, rather than the end of the sentence.

Response 8: Done.

Point 9: Consistency with reference to the 'Knowledge:pain management' in italics or not would improve readability.

Response 9: We used the term Knowledge: Pain management, in italics, along the entire article, as the name of this nursing outcome as developed in the Nursing Outcome Classification.

Point 10: Would a table to describe P4L148 improve readability?

Response 10: Thanks for this comment. We have created a table with the information about the five questions to patients, in order to make it clearer.

Point 11: The tables presented in the word columns ie Table 1 indicators; Table 2 variables; Table 3 questions; Table 4 overall score; Table 5 no heading but content below; Table 6 overall score  need to be left justified to improve readability. 

Response 11: We have checked the tables for readability and justified to the centre (because this is the style include in the template of the journal). All tables have been made using the IJERPH template.

Point 12: In 2.6, precise detail regarding what spreadsheet was used and what version of SPSS as per standard reporting would be useful.

Response 12: We have added the information about the software used.

Point 13: The first paragraph of the discussion P11L317 onwards is repetitious.

Response 13: We have revised and modified the first paragraph of the Discussion, in order to avoid repetition. The aim of this paragraph is to provide readers with a general overview of the study.

Point 14: The process of validation provides logical sequencing, however, due to the presentation of the tables, it was hard to read and comprehend. I suggest the journal table specifications are checked and complied with to improve readability

Response 14: We have checked the format of the tables using the template of the journal.

Reviewer 2 Report

I would like to thank you for the opportunity to review this very interesting manuscript. The authors have done a tremendous work. The methodology is very robust, even though there are some limitations, especially in sample selection, it seems that they do not alter the results and conclusions of the study. Many congratulations to the research team.

Author Response

Response to Reviewer 2 Comments

Point 1: I would like to thank you for the opportunity to review this very interesting manuscript. The authors have done a tremendous work. The methodology is very robust, even though there are some limitations, especially in sample selection, it seems that they do not alter the results and conclusions of the study. Many congratulations to the research team.

Response 1: Thank you for your review and comments. We agree there are some limitations in this study, we have tried to consider in the Discussion section.

Reviewer 3 Report

Greetings. This is an important and well-written article. My comments will be largely about unanswered questions I would have as a pain-management professional.

The title will be confusing to people unfamiliar with the KPM, as it appears the studies were about pain management rather than measurement of pain. Perhaps: "Validity of the KPM as a psychometric measure of acute and chronic physical pain"  The introduction is largely about pharmaceutical treatment of pain, an unfortunate limitation for several reasons. First, the implication is that historical treatment of pain has relied both on medication and outside experts; this begs the question, Why not add research regarding how pain patients have already been treating themselves so that the word "empowerment" (line 51) has background? For example, it is already known in the literature that physical exercise (wisely added as an item to the Spanish form), better eating habits, stress management, and various other practices from energy medicine (practiced in Spain) can improve upon or even eliminate pain. Second, the implication is that pain is due mostly if not entirely an organic or physical or medical causes, whereas psychotherapy innovations such as EMDR and energy psychology (practiced in Spain) have produced many case studies where chronic pain treated as a memory can often be eliminated. And thirdly, the focus on managing or living with pain, rather than healing pain, is unnecessarily pessimistic. Lines 126 onward are excellent as they clearly describe the cultural sensitivity of the researchers. The terms "bio-psycho-social", and "physical exercise" deserve elaboration; I repeat my comment about the use of "empowerment", which, I suggest, can be much more expansively discussed, as in "what do people actually do when engaging in these activities?". I will comment on this again below.  The ad hoc question in line 152, "take care of yourself", is another example of the author's valuing empowerment, and again deserves more detail. This is again repeated in lines 195-196.  Line 236 is about back pain, a common complaint around the world, one that almost always improves with time, and one that does not usually require medication. This and related comments throughout the article are opportunities to make the point that some pain is best left to the patient's care, particularly given the opioid epidemic that results from over-reliance on the pharmaceutical industry.  Table 3 is most interesting in that improvement appears to have been greatest in areas having to do with patient empowerment (take care of yourself), whereas items reflecting dependence on the outside (satisfaction with care provided by nurses and other professionals) actually worsened from baseline to final. This seems congruent with the authors valuing of self care. Perhaps this is another opportunity to investigate this further, as in, "What is it that patients are learning about their own abilities, or perhaps their own internal pharmacy, that this measurement instrument could complement and enhance?"  Let us keep in mind that a .70 correlation still leaves about half of the variance unaccounted for, and a .60 leaves 2/3 unaccounted for. Clearly there is opportunity here to add the literature on the specific example of client empowerment called the placebo effect, also spontaneous remission, both now widely studied though not included in the literature review in this article.  In line 312 and following, this is the first time relaxation and other activities were noted. They could have been introduced in the literature review.  In line 319 and earlier you combined acute with chronic pain. Might they not deserve different approaches, and is this not worth mentioning (acute deserving medication intervention to prevent production of additional pain receptors and chronic being more likely maintained by traumatic memories which often respond completely to psychotherapeutic intervention such as EMDR and EFT; see Alexander's  2012 The hidden psychology of pain and Mark Grant's EMDR protocol for chronic pain).  Lines 341 and 342. Good idea to eliminate items for interventions unavailable in Spain. However, EMDR, EFT, acupuncture, and acupressure are all taught and practiced in Spain and case studies from all look very promising.  The limitations section deserves to be noted as carefully and thoroughly written.

Author Response

Response to Reviewer 3 Comments

Point 1: Greetings. This is an important and well-written article. My comments will be largely about unanswered questions I would have as a pain-management professional.

Response 1: We are so grateful to reviewer to this detailed review. We have tried to answer all the questions and modified the text according the comments to improve its readability.

Point 2: The title will be confusing to people unfamiliar with the KPM, as it appears the studies were about pain management rather than measurement of pain. Perhaps: "Validity of the KPM as a psychometric measure of acute and chronic physical pain"

Response 2: We are grateful for this suggestion, but perhaps this change in the title is not adequate for this article. As a matter of fact, this study is focused in a tool for measurement of the knowledge of patients about the management of their pain (not about measuring the pain itself). The title shows the type of study (psychometric evaluation), the tool and the population. Nursing outcomes are a set of indicators than nurses and other healthcare providers can use in the assessment of patients (as we explained in the Introduction). So, readers could clarify this concept when reading the article.

Point 3: The introduction is largely about pharmaceutical treatment of pain, an unfortunate limitation for several reasons. First, the implication is that historical treatment of pain has relied both on medication and outside experts; this begs the question, Why not add research regarding how pain patients have already been treating themselves so that the word "empowerment" (line 51) has background? For example, it is already known in the literature that physical exercise (wisely added as an item to the Spanish form), better eating habits, stress management, and various other practices from energy medicine (practiced in Spain) can improve upon or even eliminate pain.

Response 3: We have added a paragraph about other type of therapies that patients may used to treat their pain, with appropriate references.

Point 4: Second, the implication is that pain is due mostly if not entirely an organic or physical or medical causes, whereas psychotherapy innovations such as EMDR and energy psychology (practiced in Spain) have produced many case studies where chronic pain treated as a memory can often be eliminated.

And thirdly, the focus on managing or living with pain, rather than healing pain, is unnecessarily pessimistic.

Response 4: We mostly agree with these considerations about the causes and nature of pain. However, the focus of this study is not in treating the pain, but in how healthcare providers can measure the knowledge of patients about the pain, at a basic level.

Point 5: Lines 126 onward are excellent as they clearly describe the cultural sensitivity of the researchers. The terms "bio-psycho-social", and "physical exercise" deserve elaboration; I repeat my comment about the use of "empowerment", which, I suggest, can be much more expansively discussed, as in "what do people actually do when engaging in these activities?". I will comment on this again below.

Response 5: These terms "bio-psycho-social", and "physical exercise" refer to some indicators that nurses or other healthcare providers can explore to measure the knowledge of patients. In fact, because these indicators were included and were approved by the expert panel, we have to maintain in the list of indicators of the instrument. We agree with the reviewer that pain has a multidimensional nature that could be explored; but unfortunately this go beyond the aim of our research.

Point 6: The ad hoc question in line 152, "take care of yourself", is another example of the author's valuing empowerment, and again deserves more detail. This is again repeated in lines 195-196.  

Response 6: This question were posed to patients with the aim of have a different method to measure some of pain-related aspects, in addition to the nurse measurement using the list of indicator of the KPM, in order to evaluate validity (matrix correlation in table 6. It is true that some of this question is related with empowerment of people in managing their pain, but this was not the focus of our study.

Point 7: Line 236 is about back pain, a common complaint around the world, one that almost always improves with time, and one that does not usually require medication. This and related comments throughout the article are opportunities to make the point that some pain is best left to the patient's care, particularly given the opioid epidemic that results from over-reliance on the pharmaceutical industry.  

Response 7: Again we agree with reviewer in the role of non-pharmacological therapies have in pain management. In fact, the tool tested in this study (the nursing outcome KPM, included some indicators about this). We mentioned the percentages of patients with pain located in different anatomical regions (back and other) with the aim of characterize our sample, but the discussion about the best way to treat this pain is beyond our research.

Point 8: Table 3 is most interesting in that improvement appears to have been greatest in areas having to do with patient empowerment (take care of yourself), whereas items reflecting dependence on the outside (satisfaction with care provided by nurses and other professionals) actually worsened from baseline to final. This seems congruent with the authors valuing of self care. Perhaps this is another opportunity to investigate this further, as in, "What is it that patients are learning about their own abilities, or perhaps their own internal pharmacy, that this measurement instrument could complement and enhance?"

Response 8: These five questions to patients have the aim to obtain another measurement of patients’ knowledge and satisfaction with care in order to compare with the score obtained by nurses after using the tool KPM. The aim of our study was to obtain psychometric data on reliability and validity. We agree that the question “What is it that patients are learning about their own abilities, or perhaps their own internal pharmacy, that this measurement instrument could complement and enhance?” is interesting, but unfortunately our study was no design to answer it. We have added a sentence in this paragraph to highlight the difference in the scores obtained in the five questions.  

Point 9: Let us keep in mind that a .70 correlation still leaves about half of the variance unaccounted for, and a .60 leaves 2/3 unaccounted for.

Response 9: We have added a sentence in the discussion (4.3) to point out this fact.

Point 10: Clearly there is opportunity here to add the literature on the specific example of client empowerment called the placebo effect, also spontaneous remission, both now widely studied though not included in the literature review in this article.  In line 312 and following, this is the first time relaxation and other activities were noted. They could have been introduced in the literature review.

Response 10: Some information and references about non pharmacological measures to treat pain has been added in the introduction (relaxation and other).

Point 11: In line 319 and earlier you combined acute with chronic pain. Might they not deserve different approaches, and is this not worth mentioning (acute deserving medication intervention to prevent production of additional pain receptors and chronic being more likely maintained by traumatic memories which often respond completely to psychotherapeutic intervention such as EMDR and EFT; see Alexander's  2012 The hidden psychology of pain and Mark Grant's EMDR protocol for chronic pain).

Response 11: We have modified this paragraph at the beginning of Discussion section. We agree that acute and chronic pain has different characteristics and mechanisms. However the aim of our study was not to explore this.

Thank you for this information about psychotherapeutic approaches and the EMDR intervention to treat pain, that is very interesting. This will be useful for our future research.

Point 12: Lines 341 and 342. Good idea to eliminate items for interventions unavailable in Spain. However, EMDR, EFT, acupuncture, and acupressure are all taught and practiced in Spain and case studies from all look very promising.  

Response 12: Thank you for this comment. In the list of indicators for this nursing tool there are some non-pharmacological interventions included in the original version (the nursing outcome) or proposed by the panel of experts; however, others (such as acupuncture, acupressure) were not proposed by the experts (probably because they were not frequent in practice in this context).

Point 13: The limitations section deserves to be noted as carefully and thoroughly written.

Response 13: Thank you for your comment.
